# BACA: Superpixel Segmentation with Boundary Awareness and Content Adaptation

**Nannan Liao, Baolong Guo \*, Cheng Li** **, Hui Liu and Chaoyan Zhang**

Institute of Intelligent Control and Image Engineering, Xidian University, Xi'an 710071, China
\* Correspondence: blguo@xidian.edu.cn; Tel.: +86-130-8896-6638

**Abstract:** Superpixels could aggregate pixels with similar properties, thus reducing the number of image primitives for subsequent advanced computer vision tasks. Nevertheless, existing algorithms are not effective enough to tackle computing redundancy and inaccurate segmentation. To this end, an optimized superpixel generation framework termed Boundary Awareness and Content Adaptation (BACA) is presented. Firstly, an adaptive seed sampling method based on content complexity is proposed in the initialization stage. Different from the conventional uniform mesh initialization, it takes content differentiation into consideration to incipiently eliminate the redundancy of seed distribution. In addition to the efficient initialization strategy, this work also leverages contour prior information to strengthen the boundary adherence from whole to part. During the similarity calculation of inspecting the unlabeled pixels in the non-iterative clustering framework, a multi-feature associated measurement is put forward to ameliorate the misclassification of boundary pixels. Experimental results indicate that the two optimizations could generate a synergistic effect. The integrated BACA achieves an outstanding under-segmentation error (3.34%) on the BSD dataset over the state-of-the-art performances with a minimum number of superpixels (345). Furthermore, it is not limited to image segmentation and can be facilitated by remote sensing imaging analysis.

**Keywords:** superpixel; seed initialization; boundary awareness; content adaptation



## 1. Introduction

With the massive increase in image data, traditional segmentation algorithms based on pixel processing can no longer meet the needs of daily processing. Superpixel segmentation can deal with this problem effectively to a certain extent.

The new concept of the superpixel was first proposed in [1], which defines it as a pixel collection where the color, texture or other information of pixels is basically the same. The superpixel algorithm divides the image into different regions that contain more perceptual information by using correlation measurement based on visual features. Compared with pixel features, the superpixel is a region-based feature, which is the extraction of local information from an image and it is beneficial to the expression of image structure information. Users can segment the image into hundreds or thousands of superpixels according to their own requirements, and directly process these hundreds or thousands of superpixels, which reduces the amount of data to be processed, reduces the computational complexity, speeds up the task processing, and improves the performance of the algorithm.

A superpixel does not continue to subdivide the ordinary pixels of a pixel-level image into region-level images, forming a series of pixel sets. Instead, it divides a pixel-level image into a region-level image to form a series of pixel sets. In other words, basic information elements are abstracted by adding constraints, such as color, texture [2] and distance between pixels. Superpixels "aggregate" pixels with similar properties into a larger, more representative "element" that will serve as the basic unit for other image processing algorithms. Compared with semantic segmentation and instance segmentation, superpixel

segmentation can reduce dimensions and eliminate some abnormal pixels [3]. In fact, the traditional pixel level processing can be considered to transform into superpixel level processing. Excellent superpixel algorithms can effectively extract visual information from images and improve the efficiency of the following work. They have a high comprehensive evaluation in terms of Algorithm speed [4], object contour retention [5] and superpixel shape, and are more in line with the expected segmentation effect.

Focusing on the field of remote sensing, many applications use superpixels as processing units, which is conducive to reducing data dimension and computational complexity, thus significantly improving performance, and opening up new application scenarios for superpixels in the field of remote sensing. In 2019, Girau et al. [6] proposed a color transfer model based on superpixels, using a fast parity neighbor matching algorithm to achieve color transfer. Our team has shown in [7] that the superpixel contributes to the application of remote sensing analysis and image segmentation. Arisoy et al. [8] proposed a mixture-based superpixel segmentation and classification of SAR images. With its many advantages, the superpixel has developed rapidly in recent years in domestic and foreign institutions, and new algorithms keep emerging. The superpixel algorithm has become a key technology in the field of computer vision and application research of image analysis and understanding [9–17]. Figure 1 shows the application of some superpixels in remote sensing.

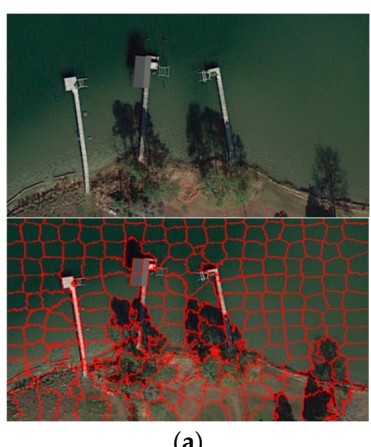 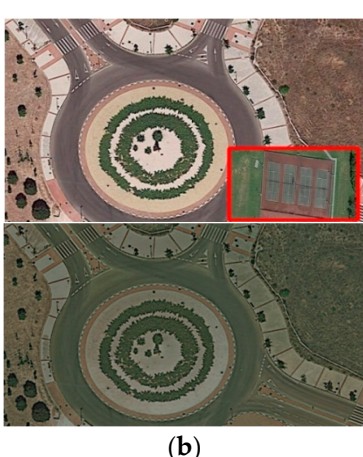 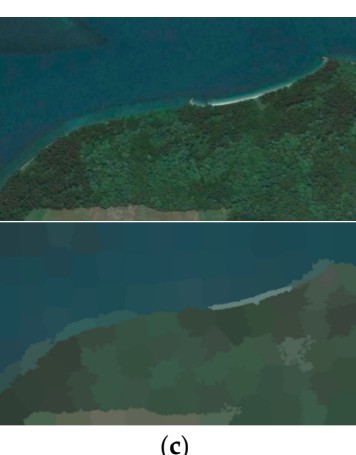

(**a**)　　　　　　　　　　　　　　　(**b**)　　　　　　　　　　　　　　　(**c**)

**Figure 1.** Three applications of superpixels in remote sensing images. (**a**) Remote sensing image segmentation; (**b**) Color transfer of remote sensing images. The image in the red box is the target image; (**c**) Reconstruction of average color on Superpixel-based is used for remote sensing tasks, such as supervised land—sea area change detection.

The existing algorithms are not effective enough to solve the problems of computational redundancy and inaccurate segmentation, which often lead to complex computation while pursuing accurate segmentation. This is contrary to the original intention of reducing computational redundancy and the number of image primitives through superpixel processing. In addition, blurred boundaries between foreground and background may occur in the image [18]. Such pixels with similar color and spatial distance are easy to be misjudged as congeneric pixels. General correlation measurements are unable to solve the above problems.

Based on these considerations, an optimized Superpixel algorithm called Superpixel Segmentation with Boundary Awareness and Content Adaptation (BACA) is proposed. In this work, we use Simple Non-iterative Clustering (SNIC) [19], a well-known and solid algorithm, which promotes the computing efficiency of Simple Linear Clustering (SLIC) [20]. Different from other superpixel algorithms whose sole purpose is to improve superpixel performance, BACA's initialization optimization and correlation measurement can be grafted onto mainstream superpixel algorithms to optimize the overall structure and improve performance.

BACA also belongs to the non-iterative clustering framework. In the initialization stage, we abandon the traditional initialization method of uniform grid sampling and generate initialization seeds adaptively through content complexity. It could distribute seeds adaptively according to the image content. Fewer seed points are allocated in the background simple region to reduce unnecessary computation and constrain the number of seeds in essence. The complex content or the region of interest formulate the initialization strategy to form detail superpixels. This is conducted to reduce the computational redundancy of pixels in a simple region.

In order to solve the problem of the non-iterative clustering framework, a large number of pixels are checked multiple times. We use the contour enhancement factor to judge the foreground and background at one time to avoid double calculation. The contour constraint metric is combined with the color-spatial five-dimensional joint metric to aggregate pixels in a more accurate manner.

In the context of previous work, the improvements and contributions of this paper can be listed as follows:

(1) An adaptive seed sampling method based on content complexity is proposed in the initialization stage, which can effectively reduce the computational cost and lays a good foundation for the subsequent steps of the superpixel algorithm.

(2) A new correlation distance measurement method integrating boundary perception and contour prior is proposed. It also overcomes the limitation of multiple calculations, further facilitating the generation of more accurate superpixels.

(3) It objectively and truly proves the feasibility of the above two points from both quantitative and qualitative aspects and compares it with the current nine excellent algorithms [19,21–28]. Experimental results further verify that BACA's segmentation results are uniform, accurate and effective.

This paper is organized as follows. Nine excellent methods are presented in the next section. In Section 3, the proposed BACA method is expanded on in detail. Qualitative and quantitative results, as well as applications in remote sensing imagesare analyzed in Section 4. Finally, Section 5 makes a brief conclusion and prospect.

## 2. Backgrounds

A large number of superpixel segmentation algorithms with remarkable performance are constantly produced in this field [29–35]; they are dedicated to being ultra-fast, high-precision, or a balanced performance to accomplish specific visual tasks. The mathematical principles of these emerging superpixel algorithms are diverse, and the classification methods in various investigations also emerge in endlessly.

1. **Accuracy-oriented.** Linear Spectral Clustering (LSC) [21] designs an approximate correlation measurement that maps pixels to a ten-dimensional feature space and then uses weighted K-means clustering to generate superpixels. It has a good output effect, but the running speed of the algorithm is slow. Entropy Rate Superpixel (ERS) [22] generates superpixels by solving the extremum of the objective function. The two terms of the objective function affect the compactness and regularity of superpixels, respectively. Therefore, the output results of this algorithm have a good boundary fit.

2. **Efficiency-oriented.** Compact Watershed (CW) [23] is an extremely efficient superpixel generating algorithm based on the marker-controlled watershed transformation. It introduces spatial constraint into the gradient-based region-growing framework, thus producing a uniform appearance with desirable segmentation quality. Superpixels extracted via energy-driven sampling (SEEDS) [24] starts from a regular grid, and then refines superpixels by constantly modifying the boundary. During the iterations, it adopts hill-climbing to solve the maximized energy cost function. In 2021, Serge Bobbia et al. proposed the Iterative Boundaries Implicit Identification (IBIS) [25] algorithm, which uses only a fraction of pixels in the image and implicitly identifies superpixel boundaries, significantly improving the computational efficiency. Xia Ren et al. proposed Structure-sensitive Superpixel Algorithm based on Non-iteration

(SSAN) [26]. By using the priority queue structure to extend the pixel label and designing a new centroid splitting and merging operator according to the manifold space area element, the structure-sensitive superpixels are quickly generated.

3. **Balance-oriented.** Achanta et al. put forward the epoch-making Simple Linear Clustering (SLIC) [20] and then updated it to Simple Non-Iterative Clustering (SNIC) [19]. Compared with the conventional K-means clustering method, it restricts the searching range and proposes a novel color-spatial distance measurement from seed points. Nevertheless, it does not consider the global information of the image due to its simplicity. In the subsequent SNIC, the iterative clustering framework is substituted by a non-iterative implementation. The optimized algorithm could execute in a single loop with better region connectivity, less memory and faster speed. Moreover, Edge Augmented Mean Shift (EAMS) [27] could search for patterns according to the density in the image, which proves sufficient boundary compliance for superpixel generation from the perspective of density estimation. Minimum Barrier Distance for Superpixel Segmentation (MBS) [28] was published in 2018 to provide a propagation scheme for clustering centers between adjacent levels on a hierarchical architecture, which makes a simple trade-off between performance and efficiency.

BACA proposed in this paper belongs to the SLIC-like framework, and this clustering superpixel algorithm is mainly introduced here.

SLIC generates superpixels by clustering pixels based on K-means clustering. In CIELAB color space, the spatial position of pixels $I_i$ in an image $I$ can be represented as vector $P(I_i) = [x(I_i), y(I_i)]$, and their color information can be represented as vector $C(I_i) = [L(I_i), a(I_i), b(I_i)]$. $L()$, $a()$ and $b()$ are the color components corresponding to pixels of images in the CLELAB color space model. $L()$ represents brightness, $a()$ is range from magenta to green, and $b()$ represents the yellow to blue range. According to the Euclidean distance between the cluster center and the pixels in the restricted region, the correlation measurement is carried out, and then the labels are assigned to each pixel. The calculation methods of color and spatial feature are Formulas (1) and (2).

$$D_c(I_i, I_j) = \|C(I_i) - C(I_j)\|_2 = \left( (L(I_i) - L(I_j))^2 + (a(I_i) - a(I_j))^2 + (b(I_i) - b(I_j))^2 \right)^{\frac{1}{2}} \tag{1}$$

$$D_s(I_i, I_j) = \|P(I_i) - P(I_j)\|_2 = \left( (x(I_i) - x(I_j))^2 + (y(I_i) - y(I_j))^2 \right)^{\frac{1}{2}} \tag{2}$$

In order to combine the above two formulas, normalization is required, and the normalized calculation method is Formula (3).

$$D(I_i, I_j) = = \left( \left( \frac{D_c}{m} \right)^2 + \left( \frac{D_s}{S} \right)^2 \right)^{\frac{1}{2}} \tag{3}$$

Among them, the values of $m$ and $S$ reflect the importance of color and spatial position characteristics to similarity, respectively. Generally, $m$ is selected as a fixed constant to measure the importance of spatial and color characteristic information. When $m$ is very large, it means that the spatial distance is more important, and when m is very small, it means that the color distance is more important. The size of $m$ affects the compactness of superpixels and it is also known as the compactness coefficient. $S$ is set to be $\sqrt{N/K}$. $N$ is the total number of image pixels, and $K$ is the number of superpixels preset by the user.

SNIC is an improvement on SLIC, which introduces the priority queue structure and shift the clustering mode from the original iterative clustering to non-iterative clustering, greatly improving the efficiency of the algorithm. The algorithm still measures the similarity between pixels in a five-dimensional Euclidean space. Distance is measured in the same way as the SLIC. The specific steps of the SNIC algorithm are as follows (Algorithm 1):

---

**Algorithm 1.** SNIC segmentation algorithm

---

**Input:** the RGB image $I$, the total number of pixels $N$, the expected number $K$, compactness $C$
**Output:** Assigned label map $L$
/*Initialization*/
Initialize the label map $L(x_i, y_i) = 0, i = 1, 2, \ldots, N$
divided the whole image into grids
Convert image $I$ from RGB space to CIELAB space
/*Joint assignment and updating*/
The centers of grids are taken as the initial clustering centers
$S_i = (L_i, a_i, b_i, x_i, y_i), i = 1, 2, \ldots, K\}$
**for** $k \in \{1, 2, \ldots, K\}$ **do**
    element $e = (x, y, k, d)$
    Push element $e$ into priority queue $Q$
**end for**
**While** $Q$ is not empty **do**
    Pop the top element $e_i$ of priority queue $Q$
    Update clustering center of all region $S_i$
    **If** $L(x_i, y_i) = 0$ **then**
      $L(x_i, y_i) = k_i$
      **for** Pop four or eight neighborhood pixels $L(x_j, y_j)$ of the pixel **do**
        Calculate the distance between pop pixel and clustering center $S_i$
        **If** $L(x_j, y_j) = 0$ **then**
          Push element $e = (x, y, k, d)$ into priority queue $Q$
        **end if**
      **end for**
    **end if**
**end while**
return Assigned label map $L$

---

## 3. Our Approach

This section introduces the proposed BACA framework in detail. Firstly, an adaptive seed sampling method based on content complexity is proposed to form an efficient initialization strategy thus avoiding seed redundancy caused by grid sampling. At the same time, the contour prior information is introduced to activate boundary awareness, which strengthens the ability of accurate boundary segmentation. The schematic of the BACA algorithm is illustrated in Figure 2.

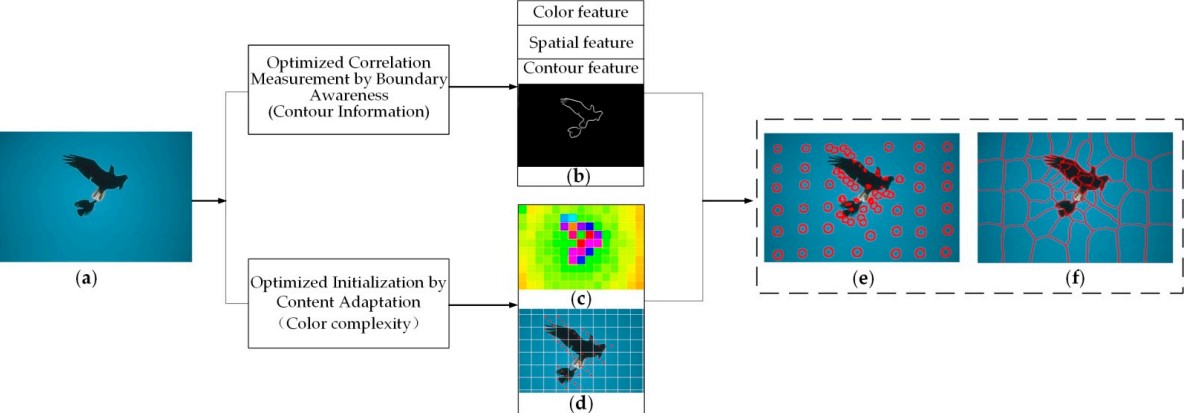

**Figure 2.** Schematic illustration of BACA superpixel generation framework. (**a**) Input image; (**b**) Contour map prior; (**c**) Diagram of color intensity; (**d**) Optimized seeds initialization by content complexity, wherein the complexity of content is represented by color feature; (**e**) The intermediate process of region growth; (**f**) Result of BACA superpixels.

### 3.1. Optimized Initialization by Complexity

In a seed-demand algorithm, the incipient location of each seed is critical to the subsequent generation of a superpixel. Nevertheless, most superpixel algorithms ignore the importance of initialization, including SLIC and its variant algorithms, which merely utilize a simple clustering (greedy) algorithm. Initially, the seeds are spread evenly over the entire image. As the steps are iterated, the seed pixels merge with the surrounding pixels to form a superpixel, as shown in Figure 3a. Figure 3b shows the initialization of SEEDS, which is to evenly divide the image into quite a few rectangles. The initial superpixels are these rectangles. With each iteration, the edges of the superpixels change until they converge.

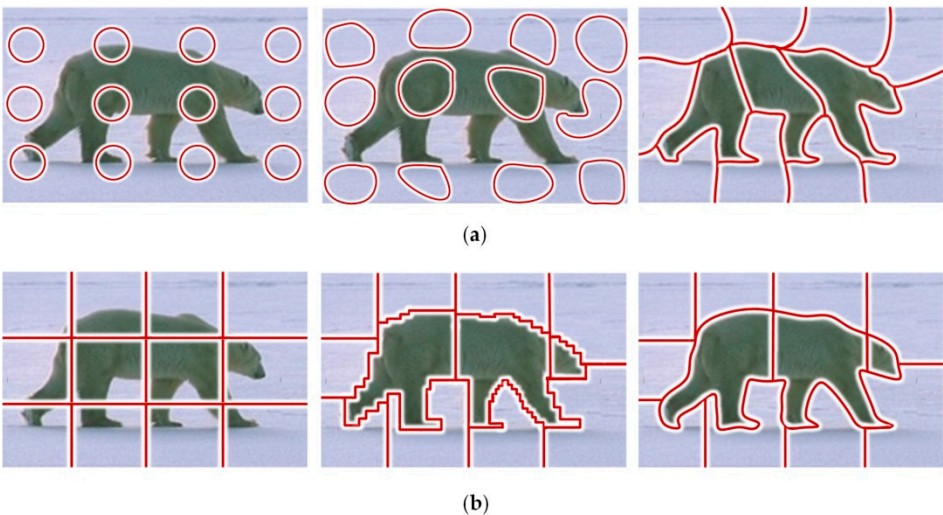

**(a)**

**(b)**

**Figure 3.** Two of the most frequently used initialization methods. (**a**) The initialization of SLIC [20] and its variant algorithms; (**b**) The initialization of SEEDS [24].

Neither of these initialization methods takes into account the complexity of the content of the image itself. In general, the content of a natural image is usually uneven, and there are multiple objects with different feature complexity. Not treating them differently incurs additional computation and time costs. For example, superpixels find it difficult to accurately capture boundaries due to their compactness. In the area of complex image content, more attention should be paid and more seeds should be set, so that the segmentation effect will be more accurate. In simple areas of image content, the number of seeds can be appropriately reduced, and then the number of superpixels can be reduced to achieve higher computational speed.

Instead of directly uniform distribution, the initialization method in this paper is based on adaptive image content. The schematic diagram of the proposed and traditional grid initialization is shown in Figure 4 below.

As the initial number of seeds increases, the Gaussian distribution can describe the color content of each superpixel [36]. As shown below:

$$f(E; \mu, \sigma) = \frac{1}{\sqrt{2\pi}\sigma} \exp\left(-\sqrt{\frac{(E-\mu)^{\mathrm{T}}(E-\mu)}{\sigma^2}}\right) \tag{4}$$

where $E$ is the color vector of $R',G',B'$ three-channel of the image, where $R' = R - G$, $G' = R + G$, $B' = \frac{1}{2}G' - B$. $\mu$ and $\sigma$ are their corresponding mean and standard deviation, respectively.

In this paper, the degree of prosperity color richness is used as a condition for image content adaptation. Color richness can measure the complexity of the image content and determine the location of seed initialization. The core of the algorithm judges the complexity of the current region and the strategy of seed initialization. The goal of the first step is to

calculate the color richness $C(i)$ of each area, the color richness $C_{\text{all}}$ of the whole image, and the mean value $\overline{C}$ of the color richness of $n$ grid areas. The calculation method of color richness is as follows:

$$C(i) = \sqrt{\sigma_{iR'}{}^2 + \sigma_{iB'}{}^2} + \varepsilon \times \sqrt{\mu_{iR'}{}^2 + \mu_{iB'}{}^2} \tag{5}$$

$$C_{all} = \sqrt{\sigma_{allR'}{}^2 + \sigma_{allB'}{}^2} + \varepsilon \times \sqrt{\mu_{allR'}{}^2 + \mu_{allB'}{}^2} \tag{6}$$

$$\overline{C} = \frac{1}{n} \sum_{i=1}^{i=n} C(i) \tag{7}$$

where $\varepsilon$ is the weight parameter set by the user. $\sigma_{R'}, \sigma_{B'}$ are the standard deviation of $R', B'$, respectively. $\mu_{R'}, \mu_{B'}$ are the mean values of $R', B'$.

$$\sigma_{R'} = \sqrt{\frac{\sum\limits_{i=1}^{n} \left(R'_i - \overline{R'}\right)^2}{n}} \tag{8}$$

$$\sigma_{B'} = \sqrt{\frac{\sum\limits_{i=1}^{n} \left(B'_i - \overline{B'}\right)^2}{n}} \tag{9}$$

$$\mu_{R'} = \frac{1}{n} \sum_{i=1}^{i=n} R'_i \tag{10}$$

$$\mu_{B'} = \frac{1}{n} \sum_{i=1}^{i=n} B'_i \tag{11}$$

where $n$ is the expected number of superpixel blocks set by the user in advance.

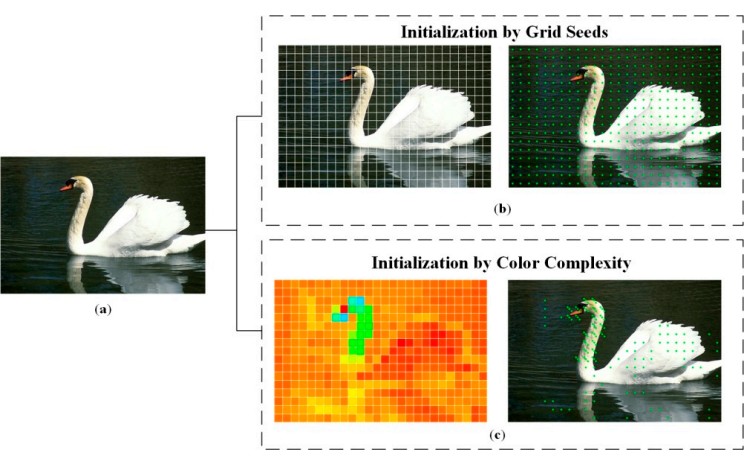

**Figure 4.** Two different seed initialization methods. (**a**) Input image; (**b**) Grid uniform seed initialization; (**c**) Seed initialization based on content adaptation. The color grid in (**c**) is a diagram of content complexity. Different colors represent different levels of complexity.

In order to make full use of the characteristics of the image itself for accurate segmentation, in the following steps, the corresponding region will be initialized adaptively according to the calculation results. After several experiments, the seed initialization strategy will be finally set up. A schematic of the content complexity is shown in Figure 5.

$$\begin{cases} C_{\max} = C_{\text{all}}, C_{\min} = \overline{C} & \text{if } \overline{C} \leq C_{\text{all}} \\ C_{\max} = \overline{C}, C_{\min} = C_{\text{all}} & \text{otherwise} \end{cases} . \tag{12}$$

1.  If the color richness of the current grid $C(i) < C_{\min}$, the current grid is defined as the content simple region;
2.  If the color richness of the current grid $C_{\min} < C(i) < C_{\max}$, the current grid is defined as the content of the general complex region;
3.  If the color richness of the current grid $C(i) \geq C_{\max}$, the current grid is defined as the content complex region.

It is worth mentioning that the natural image will appear in the whole grid, whose color is black or white, resulting in complex content but low color richness. The current region is treated in the same way as a region with general content complexity.

The pseudocode summary of initialization seed by complexity optimization is presented in Algorithm 2.

---

**Algorithm 2. Seed initialization**

---

**Input:** the RGB image $I$, the expected number $K$
**Output:** coordinates of seeds
/*Initialization*/
divided the whole image into grids
calculate the colorfulness $C_{\mathrm{all}}$ of the image $I$ by Equation (3).
**for** each cluster region $n$ **do**
    calculate the colorfulness $C(i)$ of cluster region by Equation (2).
**end for**
    calculate the mean value $\overline{C}$ of all $n$
    **for** each cluster region $n$ **do**
        **If** $C(i) \geq \max\{\overline{C}, C_{\mathrm{all}}\}$ **then**
            place three seeds evenly diagonally on cluster region $N$
        **else if** $\min\{\overline{C}, C_{\mathrm{all}}\} < C(i) < \max\{\overline{C}, C_{\mathrm{all}}\}$ **then**
            place a seed in the cluster region $N$ center
        **else** $C(i) <$ threshold
            place a seed in the cluster region $N$ center
        **end if**
    **end for**
    return coordinates of seeds

---

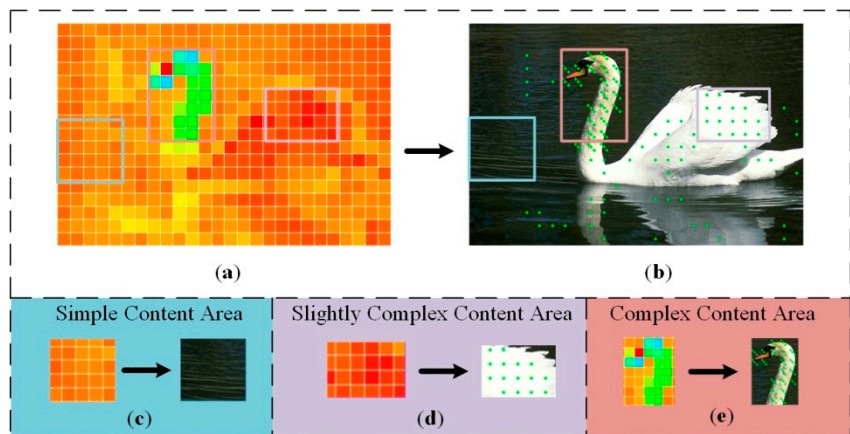

**Figure 5.** Region division of different content complexity. (**a**) Three regions with different complexity are represented by three color boxes, respectively; (**b**) Different complexity of seed initialization methods are different; (**c**) Simple content of the seed initialization method; (**d**) Slightly complex region of the seed initialization method; (**e**) Complex content of the seed initialization method.

### 3.2. Optimized Correlation Measurement

In addition to the seed initialization, the distance measurement directly impacts the clustering results. Previous works mainly adopt a joint measurement of color and spatial

position to perceive color homogeneity and reflect spatial relations, which is beneficial to the aggregation of pixels with similar color and distance in the image.

In images, the boundary between the foreground and background is frequently blurred. Seeing Figure 6, the target pixel is similar to the background pixel. In other words, the target pixel is semblable in color and distance to the background pixel but actually belongs to different objects. The color—distance metrics alone often do not do a good job of distinguishing the contours between foreground and background. In the area with small color differences, the measurement method cannot well show the difference between pixels.

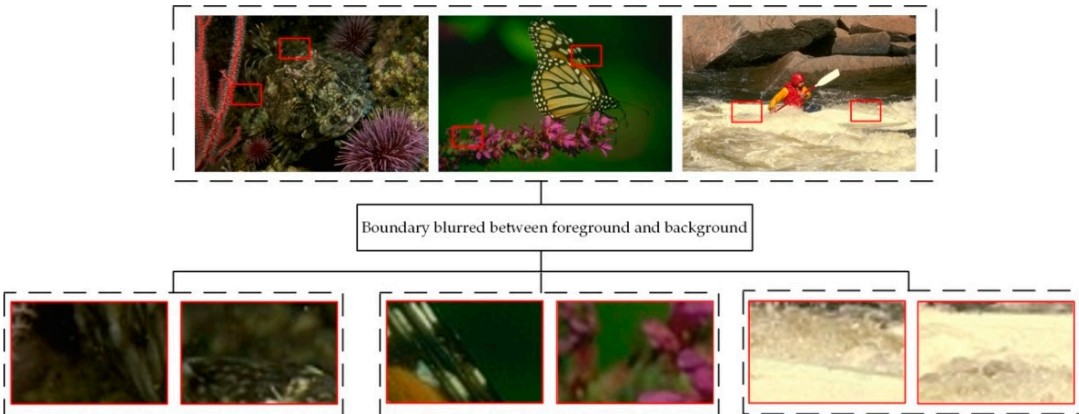

**Figure 6.** Schematic diagram of blurred boundaries between foreground and background.

It is deserved to be mentioned that the outline and edge information of foreground and background is the important information for superpixel segmentation, and also the significant basis of accurate segmentation of the superpixel. The active contour model transforms the segmentation problem into solving the minimum energy functional problem. Minima drive the contour towards the edge of the object, and eventually, it will fit perfectly. Using contour information as a tool of boundary awareness and combining it with color-distance measurement can solve the above segmentation failure problem to a certain extent and make the segmentation edges fit better.

Gray image $G = \{g_i\}_{i=1}^N$ is the contour map of image $I$ (see Figure 7d), where $g_i$ is the pixel in the contour map $G$ and $N$ is the number of pixels in the image $I$. The gray value $\phi(g_i)$ is between 0 and 255 in the contour graph $G$. If $g_i$ is a pixel on the contour line, then $\phi_{th} \leq \phi(g_i)$, $\phi_{th}$ is the preset threshold. In this paper, $\phi_{th} = 200$. Before calculating the similarity between the pixel $I_i$ and $I_j$ in an image, the linear path $L_{g_{ij}}$ between $g_i$ and $g_j$ in the corresponding contour graph $G$ is traversed to see whether there is a contour line. If there is $\phi_{th} \leq \phi(g_k)$, $g_k \in L_{g_{ij}}$ on the linear path $L_{g_{ij}}$ between $g_i$ and $g_j$, it indicates that there are contours between pixels $I_i$ and $I_j$ in image $I$, as shown in Figure 7.

$$\lambda(I_i, I_j) = \begin{cases} 1, & \text{if } \exists g_k \in L_{g_{ij}}, \text{s.t. } \phi_{th} \leq \phi(g_k) \\ 0, & \text{if } \forall g_k \in L_{g_{ij}}, \text{s.t. } \phi_{th} > \phi(g_k) \end{cases} \tag{13}$$

Combining the contour constraint metric with the color-distance five-dimensional joint metric:

$$D_{new} = D(I_i, I_j) \cdot (1 + \omega \cdot \lambda(I_i, I_j)) \tag{14}$$

where $\omega$ is the contour enhancement factor. When $\lambda(I_i, I_j) = 0$, the formula is the color-space five-dimensional joint metric.

After the optimization of the color-spatial five-dimensional joint metric, the image contour information is reflected in the formula as an important impact factor. When the difference between foreground and background is small, the trimmer as the contour

enhancement factor will play an important role in pixel classification. The correlation measurement framework incorporating contour information is shown in Figure 7.

It should be noted here that any effective contour edge detection algorithm can be applied to the framework of this paper. The choice of a suitable contour detection algorithm depends on the current application scenarios and fields.

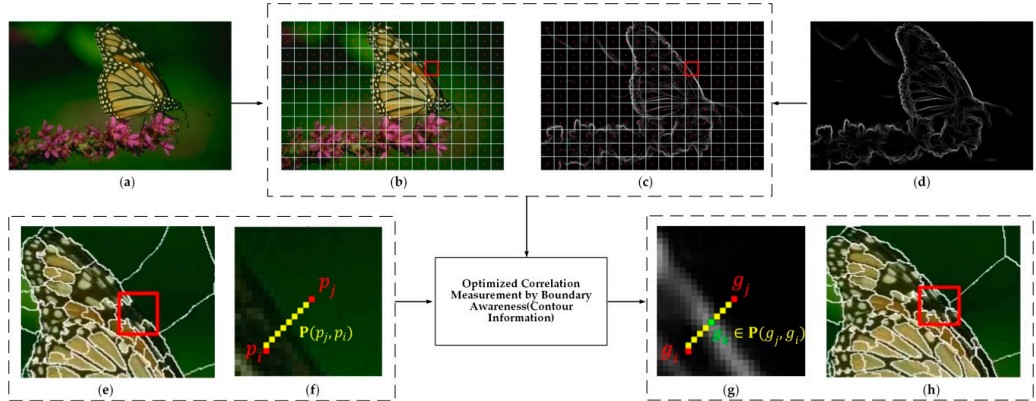

**Figure 7.** Workflow of superpixels generated by the optimized correlation measurement. (**a**) Input image; (**b**,**c**) Schematic diagrams of initial seed distribution of (**a**,**d**); (**d**) Contour prior of (**a**); (**e**) Result of superpixel generation without contour information; (**f**) Illustration of pixels in different context in input image; (**g**) Illustration of pixels in different context in contour gray image; (**h**) Final result of superpixel generation.

## 4. Experiment and Discussion

The experimental and analytical framework is designed. First, the dataset and the related algorithms for comparison are explained. Secondly, the qualitative and quantitative advantages of BACA are evaluated from whole to part, and the synergistic effect of the two strategies is further verified by ablation experiments. More importantly, BACA is discussed in the field of remote sensing in Section 4.3.

### 4.1. Experiment Setup

This experiment is performed on the Berkeley Segmentation Data Set and Benchmarks 500 (BSDS500) [37], including 500 natural images, ground-truth human annotations and benchmarking code. In this paper, 150 images are randomly selected in the data set for the experiment. Accuracy-oriented superpixel algorithms LSC [21], ERS [22], efficiency-oriented superpixel algorithms CW [23], SEEDS [24], IBIS [25], SSAN [26] and balance-oriented superpixel algorithms SNIC [19], EAMS [27], IBIS [28] are compared to prove the superiority of BACA. The abovementioned methods borrow their default parameters and code. All methods are executed on an Intel Core i7 4.2 GHz with 16 GB RAM without any parallelization or GPU processing.

### 4.2. Algorithm Analysis

In this part, the performance of the proposed BACA is fully verified and analyzed to verify its superiority. Firstly, the visual effects of the output superpixel are demonstrated along with several other state-of-the-art (SOTA) algorithms, which are all based on valid code and parameters. Secondly, the quantitative results of several metrics including Boundary Recall (BR), Achievable Segmentation Accuracy (ASA), Under-segmentation Error (UE) and Compactness (CO) are compared to show the desirable performance. Finally, the effectiveness of boundary awareness for the superpixel algorithm is verified by ablation experiments.

### 4.2.1. Visual Comparisons of Superpixel Results

Figure 8 shows the visual segmentation results of eight methods on the BSDS500 dataset. It can be observed from Figure 8c,d that the segmentation curves of EAMS, ERS and SEEDS are chaotic and uneven. Among them is ERS, which captures almost all the details, but the over-segmentation results cannot provide support for successful work. On the contrary, the remaining five algorithms could generate compact and uniform superpixels. CW is a spatially constrained superpixel algorithm with a controllable number and compactness to produce a trim appearance. However, BACA was better able to segment the contours of objects in areas with complex colors. LSC keeps the segmentation relatively accurate but the anti-texture effect is dissatisfactory. SNIC passably balances shape regularity and boundary adherence. In particular, the BACA algorithm can adaptively adjust the granularity of segmentation according to the contour structure and color of different regions in the image.

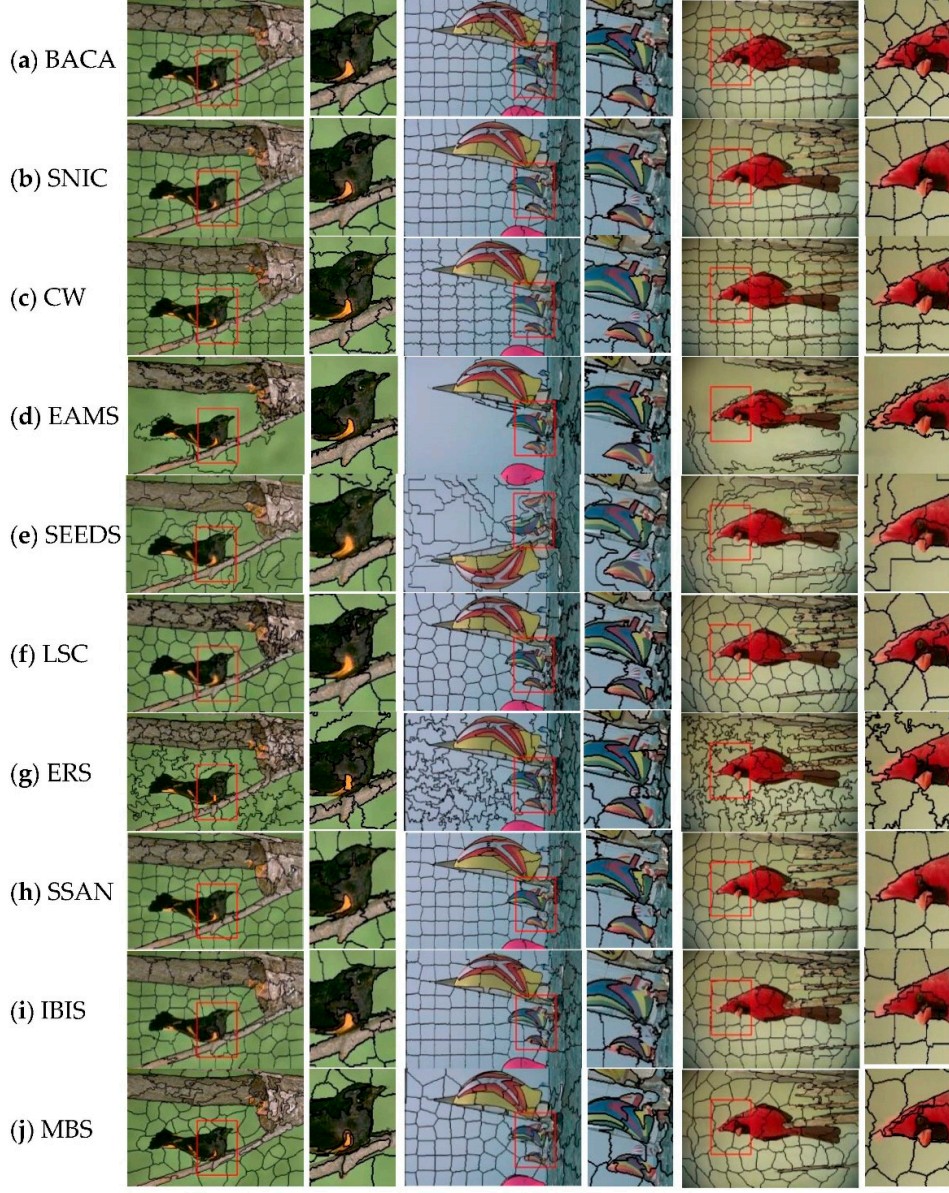

**Figure 8.** Visual comparison of segmentation results with 100 expected superpixels. (**a**) BACA; (**b**) SNIC; (**c**) CW; (**d**) EAMS; (**e**) SEEDS; (**f**) LSC; (**g**) ERS; (**h**) SSAN; (**i**) IBIS; (**j**) MBS. Alternating columns show each segmented image followed by the zoom-in performance.

### 4.2.2. Quantitative Evaluation by Metrics

In addition to the visual evaluation, four evaluation indicators are introduced in this section for quantitative and objective analysis [38]:

1. Boundary Recall (BR). BR is a popular metric to describe the fit degree between the superpixel outlines and the object boundaries. A greater BR indicates that the superpixel boundary is closer to the real boundary of the image.
2. Under-segmentation Error (UE). UE measures a ratio of the spilled pixels to the real segmented pixels, wherein the former refers to the pixels beyond the intersection of the superpixel and the ground truth. It is negatively correlated with segmentation accuracy.
3. Achievable Segmentation Accuracy (ASA). ASA describes the accuracy of segmentation results. It reveals the percentage of the correct segmentation in terms of the ground truth.
4. Compactness (CO). CO describes the roundness of each superpixel block, which is positively associated with the regularity and uniformity of the superpixel shape.

Figure 9 gives a quantitative analysis of eight methods using the above four indicators. Theoretically speaking, BR measures the fitting rate between the segmentation result boundary and the Ground Truth boundary. From a practical point of view, some algorithms with a high BR value have over-segmentation, resulting in meandering and irregular segmentation curves, such as ERS and SEEDS. Figure 9d measures the compactness and regularity of the superpixel block, and CO is another important index to measure the performance of the superpixel algorithm. The CO values in ERS and SEEDS further confirmed the presence of false detections. The ASA best illustrates the effectiveness of the superpixel algorithm. The inclusion of contour information gives BACA a comparable performance to ASA. When k = 50, BACA's UE value is 24% lower than the best performer among the remaining SOTA algorithms and more than two times lower than the most unsatisfactory (the lower the UE value, the better). BACA is the optimization product of SNIC. Compared with the SNIC algorithm, BACA has advantages no matter which of the four indicators is considered.

### 4.2.3. Ablation Experiments

As the name suggests, BACA contains two optimizations, boundary awareness and content adaptation, which work synergistically on the clustering framework. In order to further illustrate the effectiveness of every single strategy on SNIC, BA-SNIC (boundary aware SNIC) and CA-SNIC (content adapted SNIC) are designed to be the baselines, respectively. This section further explores the impact of these two strategies on the overall framework from both qualitative and quantitative aspects. In addition, the relationships and differences among SNIC, BA-SNIC, CA-SNIC and BACA are also explained.

Figure 10 visually shows the qualitative analysis results of SNIC, BA-SNIC, CA-SNIC and BACA. Compared with the conventional SNIC, BA-SNIC adds contour terms to measure the similarity, and the effects are quite immediate. For example, SNIC sometimes cannot accurately distinguish the boundary between foreground and background in the area with a small color difference between foreground and background. Conversely, BA-SNIC could accurately segment regions with small color differences based on prior contours. Different from the SNIC algorithm, the CA-SNIC algorithm makes positive improvements in seed initialization. The seed initialization within CA-SNIC could adaptively adjust the number of seeds in different regions. It places more seeds in the region with complex color and structure to produce finer segmentation, which essentially promotes more detailed boundary adherence. BACA combines the advantages of BA-SNIC and CA-SNIC to achieve more comprehensive superpixel segmentation results.

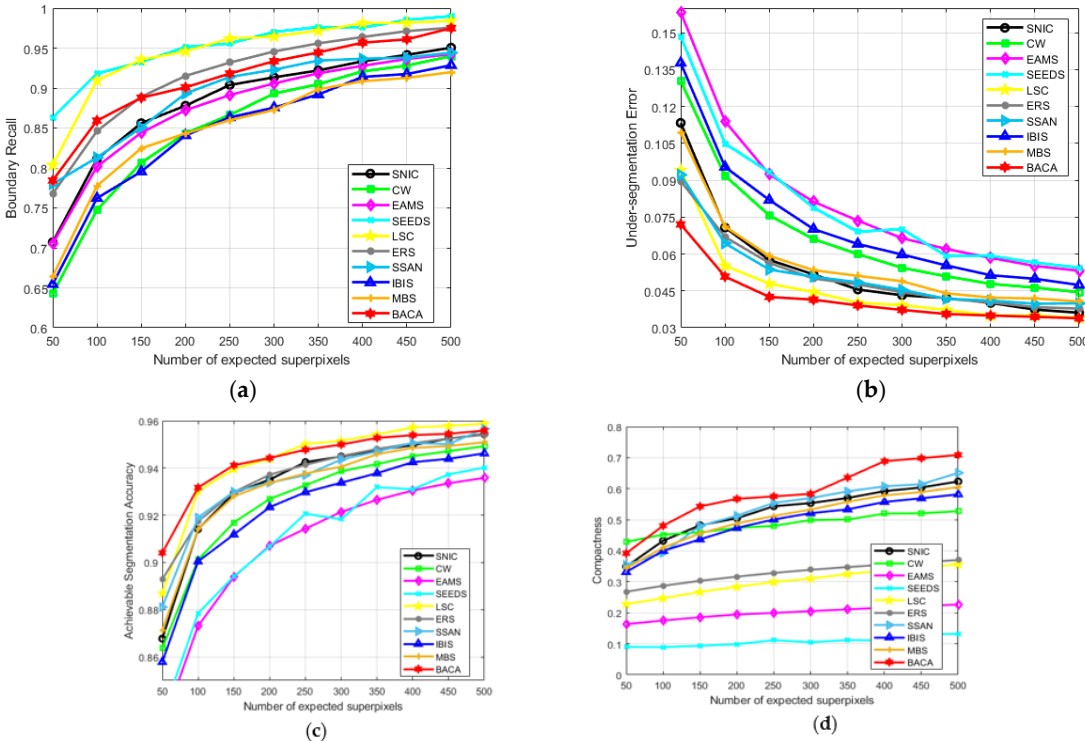

**Figure 9.** Quantitative evaluation of different algorithms on four evaluation indicators. (**a**) Boundary recall; (**b**) Under-segmentation error; (**c**) Achievable segmentation accuracy; (**d**) Compactness. The expected number of superpixels ranges from 50 to 500.

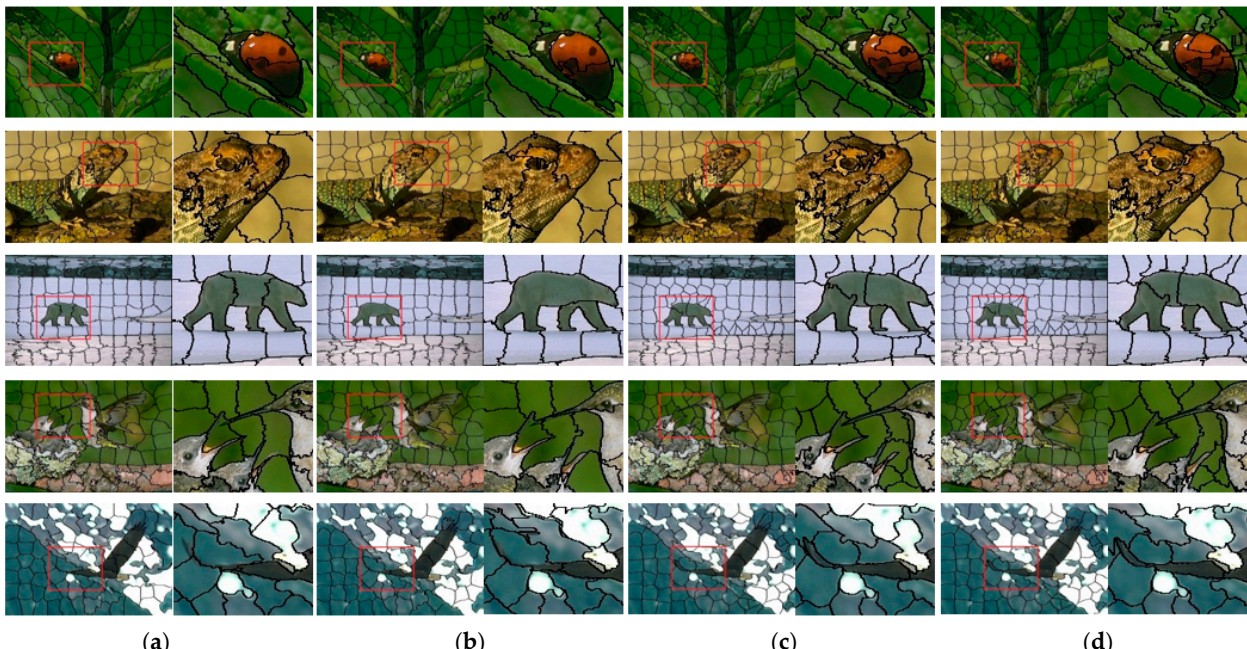

**Figure 10.** Visual results of superpixels on BSDS500. Each column represents superpixels generated by (**a**) SNIC; (**b**) BA-SNIC; (**c**) CA-SNIC; (**d**) BACA. The excepted number of superpixels is fixed to 100. Alternating columns show each segmented image followed by local details.

Figure 11 illustrates the quantitative results from four aspects. In comparison to SNIC, CA-SNIC increases the content complexity in the initialization stage, which adaptively

adjusts the initialization according to the complexity of the image itself. As a result, it could avoid seed redundancy, thus reducing the computational complexity from the beginning. For BA-SNIC, contour information makes the segmentation curve closer to the boundary and performs better on UE and ASA. Meanwhile, the content adaptive optimization step improves the compactness of the superpixel block, resulting in a better visual effect.

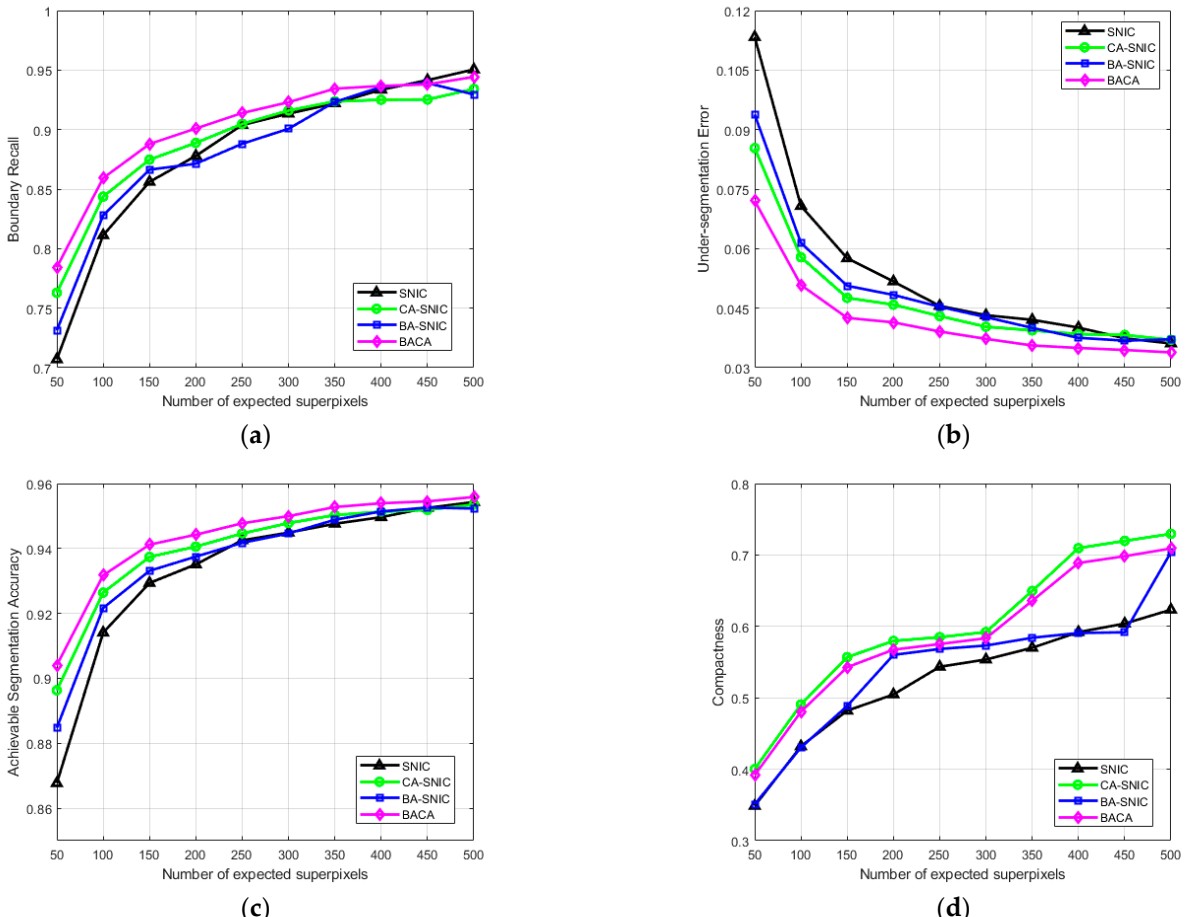

**Figure 11.** Quantitative evaluation of different algorithms on four evaluation indicators. (**a**) Boundary recall; (**b**) Under-segmentation error; (**c**) Achievable segmentation accuracy; (**d**) Compactness. The expected number of superpixels ranges from 50 to 500.

In addition to the above four indicators, the actual and expected segmentation numbers of superpixels are also the focus of the researcher. The number of superpixels determines the refinement degree of the segmentation effect. The more the number of superpixels, the more redundancy and over-segmentation will occur to a certain extent. Content adaptation can precisely solve this problem, ensuring that BACA has the minimum number of actual superpixels while holding the same performance as SOTA. As shown in Table 1.

**Table 1.** Comparison of the superpixels number between actual generation and user-expected on the eleven algorithms.

| Algorithm | Expected Superpixel Number | | | | | | | | | |
|---|---|---|---|---|---|---|---|---|---|---|
| | 50 | 100 | 150 | 200 | 250 | 300 | 350 | 400 | 450 | 500 |
| SLIC | 40 | 94 | 146 | 185 | 226 | 260 | 327 | 378 | 397 | 439 |
| SNIC | 40 | 96 | 150 | 187 | 260 | 294 | 330 | 400 | 442 | 504 |
| CW | 50 | 101 | 145 | 198 | 242 | 309 | 346 | 407 | 447 | 509 |
| EAMS | 50 | 100 | 150 | 200 | 250 | 300 | 350 | 400 | 450 | 500 |
| SEEDS | 88 | 145 | 197 | 246 | 280 | 350 | 384 | 453 | 481 | 533 |
| LSC | 76 | 141 | 205 | 279 | 343 | 413 | 470 | 562 | 658 | 758 |
| ERS | 50 | 100 | 150 | 200 | 250 | 300 | 350 | 400 | 450 | 500 |
| SSAN | 38 | 89 | 138 | 175 | 244 | 304 | 349 | 390 | 432 | 497 |
| IBIS | 40 | 93 | 125 | 182 | 223 | 256 | 291 | 372 | 392 | 435 |
| MBS | 40 | 96 | 150 | 187 | 228 | 260 | 330 | 394 | 400 | 442 |
| BACA | 38 | 87 | 130 | 158 | 188 | 211 | 262 | 302 | 314 | 345 |

*4.3. More Discussion*

With the development of remote sensing technology, the characteristics of remote image sensing, such as a large amount of data, high complexity and broad perspective, are increasingly prominent. Typical feature analysis, road extraction, urban planning and other practical applications are of great civil and military significance. The traditional segmentation algorithm can only extract low-level features, which cannot meet the requirements of high-resolution remote sensing image segmentation. In order to prove that the proposed superpixel generation algorithm BACA is beneficial to the analysis of remote sensing images, BACA is compared with the latest two published segmentation algorithms in three years in three years and the classical SNIC framework. In addition, the above sections have detailed the excellent performance of BACA in four evaluation indicators (BR, UE, ASA, CO), and this section focuses on the comparison of visual quality and the number of primitives to be processed.

In order to ensure the universality of the algorithm in different application scenarios, images from three different remote sensing datasets are used for experiments, some of which are shown in Figure 12. It is worth noting that the size of the images in each dataset is not consistent, so the size of the images should be unified to ensure the scientificity and accuracy of the experiment. There is no limit to the size of the image.

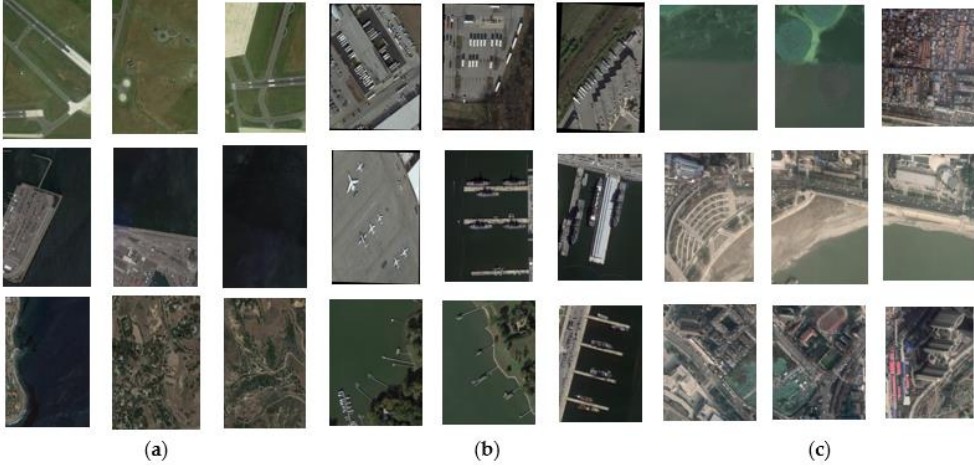

(a)  (b)  (c)

**Figure 12.** Remote sensing images of parts of the three datasets. (**a**) NWPU VHR-10 [39]; (**b**) DOTA [40]; (**c**) CHN6-CUG [41]. Each dataset is collected in different scenarios, such as airport, sea area, wharf, city road, farmland, etc.

Figure 13 shows the visual quality performance of the four algorithms, including BACA, on the three datasets. It can be seen from Figure 13a that the pixel segmentation blocks are large and sparse in the region with simple image content, while the pixel segmentation blocks are small and dense in the region with more details in the image content. This is because the initial seed points can be distributed adaptively with the addition of the content adaptive strategy. As a superpixel generation framework optimized from the overall structure, the boundary sensing strategy makes BACA superior in boundary fitting. It can also be accurately segmented small targets, as shown by the detection of small boats in Picture 3.

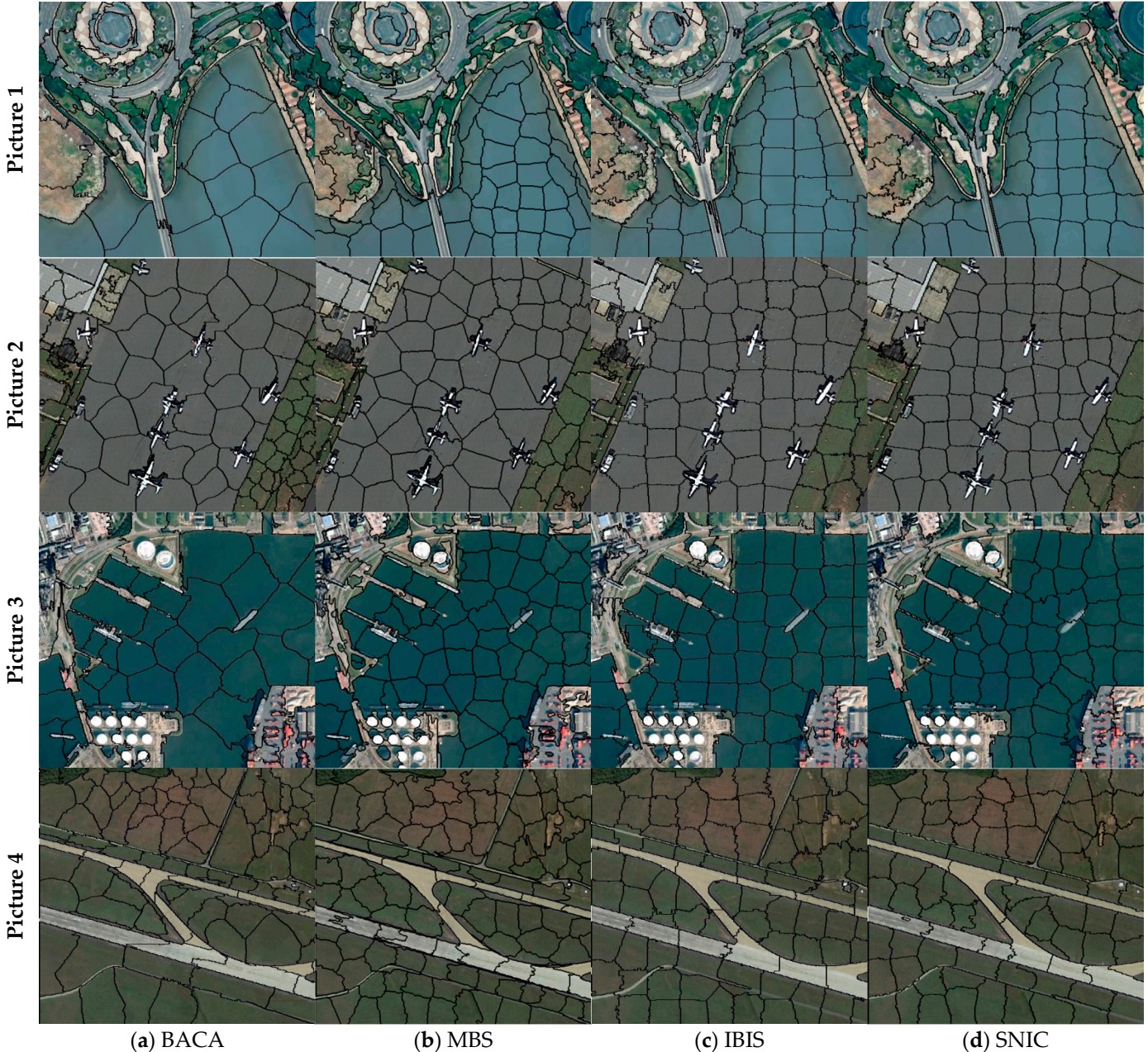

**Figure 13.** Visual comparison of segmentation results with 100 expected superpixels. (**a**) BACA; (**b**) MBS; (**c**) IBIS; (**d**) SNIC.

From another perspective, the actual generated superpixel block is the basic processing unit of subsequent visual tasks, which determines the computational amount of the subsequent processing. BACA has great advantages in actually generating the number

of superpixels and keeps the minimum amount of computation to be processed without losing or even better segmentation accuracy. Table 2 illustrates the numerical comparison of superpixels between actual generation and user-preset (k = 100) superpixels of the four images in Figure 13 under different algorithms.

**Table 2.** Actual superpixels number of different algorithms on different pictures (k = 100).

| Picture | Algorithm (k = 100) | | | |
|---|---|---|---|---|
| | **BACA** | **MBS** | **IBIS** | **SNIC** |
| Picture 1 | 75 | 96 | 90 | 90 |
| Picture 2 | 82 | 96 | 93 | 90 |
| Picture 3 | 51 | 96 | 75 | 90 |
| Picture 4 | 83 | 96 | 88 | 90 |

A total of 150 images are randomly selected from three remote sensing datasets for the experiment. The parameters are adjusted so that the segmentation accuracy has an identical accuracy level. Under this condition, the actual number of superpixels of different algorithms can be calculated with different preset numbers of superpixels. As shown in Table 3 for details.

**Table 3.** Comparison of the superpixels number between actual generation and user-preset on the four algorithms.

| Algorithm | User-Preset Superpixel Number | | | | | | | | | |
|---|---|---|---|---|---|---|---|---|---|---|
| | **50** | **100** | **150** | **200** | **250** | **300** | **350** | **400** | **450** | **500** |
| MBS | 46 | 98 | 153 | 192 | 233 | 278 | 334 | 398 | 426 | 463 |
| IBIS | 49 | 98 | 130 | 192 | 230 | 264 | 304 | 372 | 415 | 458 |
| SNIC | 40 | 93 | 150 | 196 | 274 | 298 | 336 | 400 | 436 | 402 |
| BACA | 32 | 73 | 128 | 146 | 162 | 196 | 223 | 264 | 305 | 321 |

In summary, BACA can play a positive role in the preprocessing stage of remote sensing images with a large amount of data and strong complexity, significantly reducing the number of units to be processed, fundamentally reducing the burden of subsequent visual tasks and thus contributing to the analysis and processing of remote sensing images.

**5. Conclusions**

This paper presents an optimized superpixel generation framework, termed Boundary Awareness and Content Adaptation (BACA). Firstly, a new seed initialization method is proposed with emphasis on the complexity of image content, which could both reduce the number of superpixels as well as the amount of computation. As a result, it improves the running efficiency and acquires content adaptation. In addition, a multi-feature correlation measurement is proposed to improve the misclassification of boundary pixels. The contour prior information enables the whole algorithm framework to follow the real boundary and further achieves excellent performance in segmentation accuracy. The proposed BACA achieved a significantly improved UE (3.34%) and ASA (95.59%) over the state-of-the-art performances while maintaining the highest CO (70.91%) and a minimum number of superpixels (345). Experimental results indicate that the two optimizations could generate a synergistic effect, which runs in a limited actual superpixel number with the most advanced performance on the public dataset.

Future work will focus more on exploring strategies to improve the work efficiency of BACA and applying the proposed algorithm to advanced tasks in the remote sensing field. For example, most trackers use high-level appearance structures or low-level clues to represent and match target objects. Inspired by BACA, a discriminative appearance model

based on superpixels can be proposed from the perspective of intermediate vision. The target and background in remote sensing images are quickly distinguished by a tracking method that captures the structural information in the superpixel algorithm.

**Author Contributions:** Conceptualization, methodology and writing—original draft preparation, N.L.; validation and formal analysis, C.L.; investigation and data curation, H.L.; writing—review and editing, C.Z.; project administration and funding acquisition, B.G. All authors have read and agreed to the published version of the manuscript.

**Funding:** This research is supported financially by National Natural Science Foundation of China (Grant No. 62171341).

**Institutional Review Board Statement:** Not applicable.

**Informed Consent Statement:** Not applicable.

**Data Availability Statement:** The BSDS500 dataset and the reference codes in this work are available at: https://github.com/davidstutz/superpixel-benchmark (accessed on 29 January 2021). The NWPU VHR-10 dataset and the reference codes in this work are available at: https://drive.google.com/open?id=1--foZ3dV5OCsqXQXT84UeKtrAqc5CkAE (accessed on 9 November 2014). The DOAT dataset and the reference codes in this work are available at: https://captain-whu.github.io/DOTA/index.html (accessed on 26 January 2018). The CHN6-CUG Road Dataset and the reference codes in this work are available at: http://grzy.cug.edu.cn/zhuqiqi/zh_CN/yjgk/32368/list/index.htm (accessed on 25 May 2021).

**Acknowledgments:** The authors would like to thank the editor and anonymous reviewers for their valuable comments on this paper.

**Conflicts of Interest:** The authors declare no conflict of interest.

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
