# Peer review of "BACA: Superpixel Segmentation with Boundary Awareness and Content Adaptation"

_remotesensing, doi:10.3390/rs14184572_

Round 1

Reviewer 1 Report

The paper presents an optimized superpixel generation framework. The following concerns should be considered before it can be acceptable for publication.

1. More SOTA superpixel generation methods should be included in the literature review and comparison experiments, espectially those published in recent three years.

2. I am not sure what you want to show in Figure 1b. The caption may be not suitable for its content and the surrounding text.

3. Citations for dataset, metrics, and compared methods are lost in Section 4.1 and 4.3.

4. Experiments on remote sensing images should be added to suport the claimed "facilitate remote sensing imaging analysis" in the abstract.

Reviewer 2 Report

The manuscript discussed an image segmentation method based on superpixel approach. Main contributions are 1) using a seed initialization strategy by considering the color complexity of images and 2) clustering pixels by a new correlation distance integrating boundary and contour information. Overall, the technical contributions of the manuscript are not strong. Especially, in the contribution 2, i.e., the correlation distance integrating boundary and contour information, one may argue that either the boundaries or the contours have significant differences with those textured areas. Thus, they can be easily differentiated from other pixels during the process of clustering for any ordinary correlation methods. Moreover, the effects of only using color complexity for seed initialization are also suspicious since the image complexity also includes gray levels’ information.

 The writing and organization of the manuscript can also be improved greatly.

(1). Page 1, “forming a series of pixel sets, Instead, it divides a pixel-level” -> “forming a series of pixel sets. Instead, it divides a pixel-level”

(2). Page 2, “My team has shown in [4] that superpixel contributes “-> “Our team has shown in [4] that superpixel contributes “

(3). Page 4, “a() range from magenta to green” ->  “a() is ranged from magenta to green”

(4) Page 4, “The calculation methods of color and spatial feature are formula (1) and Formula (2).” -> “The calculation methods of color and spatial feature are Formula (1) and Formula (2).”

Reviewer 3 Report

Authors contributions:

The authors have proposed a new seed initialization strategy, which can distribute seeds according to the image content. Simple region is needed to reduce unnecessary computation and constrain the number of seeds in essence.

The proposed initialization strategy can effectively reduce the computational cost while ensuring the segmentation effect, which lays a good foundation for the subsequent steps of the proposed super pixel algorithm.

A new correlation distance measurement method integrating boundary perception and contour prior is proposed.

The algorithm proposed in this paper is undisputable. It integrates effective initialization strategy and contour prior information.

I have some reviewer notes:

Abstract. You have to present your results with values, not only with text descriptions.

Introduction. The aim of this work is not clearly presented.

Lines 155 and 156. Lab, “L” have to be with capital letter. Also, you have to define if it is CIE L*a*b* or Hunter Lab.

Equation (4). If you use mean and standard deviation, you have to define what is your level of significance.

Line 317. Citation is missing [].

Line 318. How do you choose the amount of sample size? Do you have test and validation samples?

Discussion part is missing. You have to compare your results with those from minimum three other papers.

Conclusion. You have to present your results with numbers. How your results improve the known solutions in this study area? What are the limitations of your work? How the work will be continued?

I have some suggestions:

Present your results with values. Give more detailed information about the methods and technical tools used. These suggestions will improve your contribution.

Round 2

Reviewer 1 Report

Thanks for the authors' revision according to my previous comments. Current version has been significantly improved and can be acceptable.

Author Response

Dear Editor:

Thank you for your letter and the reviewers' comments concerning our manuscript entitled " BACA: Superpixel Segmentation with Boundary Awareness and Content Adaptation" (remotesensing-1883246). Those comments are all valuable and very helpful for revising and improving our paper, as well as the important guiding significance to us.

Again, thank you for your timely notification and consideration of this paper.

Best regards!

Nannan Liao

Reviewer 2 Report

The authors addressed my comments in the last reviewing. Please check the commas in Equations (6) and (7). Also Equation (8) can be re-formatted by two rows, and for Equation (9), (10), etc.
